# Motion Analysis of Balance Pre and Post Sensorimotor Exercises to Enhance Elderly Mobility: A Case Study

**Ji Chen \*, Roni Romero**  **and Lara A. Thompson**

Biomedical Engineering Program, Department of Mechanical Engineering, University of the District of Columbia, Washington, DC 20008, USA
**\*** Correspondence: ji.chen@udc.edu

**Abstract:** Quantitative assessment of movement using motion capture provides insights on mobility which are not evident from clinical evaluation. Here, in older individuals that were healthy or had suffered a stroke, we aimed to investigate their balance in terms of changes in body kinematics and muscle activity. Our research question involved determining the effects on post- compared to pre-sensorimotor training exercises on maintaining or improving balance. Our research hypothesis was that training would improve the gait and balance by increasing joint angles and extensor muscle activities in lower extremities and spatiotemporal measures of stroke and elderly people. This manuscript describes a motion capture-based evaluation protocol to assess joint angles and spatiotemporal parameters (cadence, step length and walking speed), as well as major extensor and flexor muscle activities. We also conducted a case study on a healthy older participant (male, age, 65) and an older participant with chronic stroke (female, age, 55). Both participants performed a walking task along a path with a rectangular shape which included tandem walking forward, right side stepping, tandem walking backward, left side stepping to the starting location. For the stroke participant, the training improved the task completion time by 19 s. Her impaired left leg had improved step length (by 0.197 m) and cadence (by 10 steps/min) when walking forward, and cadence (by 12 steps/min) when walking backward. The non-impaired right leg improved cadence when walking forward (by 15 steps/min) and backward (by 27 steps/min). The joint range of motion (ROM) did not change in most cases. However, the ROM of the hip joint increased significantly by 5.8 degrees ($p = 0.019$) on the left leg side whereas the ROMs of hip joint and knee joint increased significantly by 4.1 degrees ($p = 0.046$) and 8.1 degrees ($p = 0.007$) on the right leg side during backward walking. For the healthy participant, the significant changes were only found in his right knee joint ROM having increased by 4.2 degrees ($p = 0.031$) and in his left ankle joint ROM having increased by 5.5 degrees ($p = 0.006$) during the left side stepping.

**Keywords:** sensorimotor training; evaluation protocol; joint angle; spatiotemporal; muscle activities; falls; elderly; balance

## 1. Introduction

According to the Centers for Disease Control and Prevention, more than 32,000 older adults (age 65+) die from falls every year [1]. In the United States, healthcare spending on older adult falls has been approximately USD 50 billion annually [2]. Moreover, the elderly population has a higher risk of stroke [3]. Nearly half of the 6.5 million individuals who survive a stroke will have moderate to severe neurological deficits, 30% will be unable to walk unassisted and over 25% will need assistance in their daily activities [4]. These impairments lead to activity limitation, decreased independence, falls and fear of falling [5].

Sensory integration refers to a mechanism that combines orientation information (often represented as a weighted combination of sensory inputs) to serve as a basis for generating corrective actions that facilitate balance stabilization [6]. Mahoney et al. reported that magnitude of multisensory integration can predict balance performance [7]. Multisensory

integration impairment is found to be present in people with chronic stroke [8], older adults [9], and individuals with Parkinson's Disease [10]. The deficits in attentional control and a larger time window of integration may explain why the impaired integration leads to postural instability and even falls in elderly people [9]. When the elderly and other people with neurologic deficits try to maintain balance and simultaneously perform another task, the activity of the brain region associated with balance is significantly reduced while the brain regions of unrelated modalities are more active, especially in fall-prone individuals [11].

It is posited that sensorimotor training (SMT), which includes sensory conflict conditions (varying inputs to the sensory systems), will be efficient in facilitating multisensory integration and thus, in enhancing the balance of survivors of stroke and older adults [12]. Sensory conflicts in training can be achieved by distorting somatosensory inputs using soft surfaces on the ground or by restricting visual input by closing or covering the eyes, or by challenging vestibular inputs during walking backwards or in conditions where the vision and somatosensory inputs are limited or unreliable. A previous review on SMT studies highlighted Janda's theory that the proprioception serves the critical role in balance and coordinated movement, and that people progress by performing exercises in different postures, bases of support, and challenges to their center of gravity through three stages of SMT: static, dynamic and functional [13].

Ideally, the design of SMT should aim to facilitate not only the sensory input, but also integration of information in the central nervous system and motor output. In particular, a few functions and systems need to be intact in order for a person to have normal gait. These functions and systems include the locomotor function, which initiates and sustains rhythmic gait, postural reflex, sensory function, sensory integration, the musculoskeletal system and cardiopulmonary functions [14]. The tasks in our SMT focus on the proprioceptive stimulations, locomotor function, and the musculoskeletal system by asking participants to perform the standing, step-up, and walking tasks on different surfaces with stable posture through three stages. In addition, few SMT studies had developed the details of tasks that aim to improve attentional control and reduce sensory integration time. To develop these tasks is rather difficult, as heightened multisensory integration due to aging can lead to performance decrements when sensory inputs are in conflict [15]. To address this difficulty, our SMT tasks are designed to train elderly people in a safe environment to become adaptive to using reliable sensory cues, when exposed to different sensory conflicts among visual, vestibular and somatosensory systems.

The scientific question for our assessment study is the following: can our 6-week-long SMT improve the gait and balance of people with weakened sensory systems? Biomechanical analysis is critical to the question we proposed. A motion capture-based assessment pipeline was thus developed to evaluate the joint angles, muscle activities and spatiotemporal parameters. We expect that kinematic and surface EMG data are more sensitive in detecting the changes of dynamic balancing between pre and post training than commonly used clinical scales

This manuscript aims to explain our assessment protocol and, as a case study, to describe motion analysis that investigates the effects of our sensorimotor training on the mobility of two elderly participants (one with chronic stroke). Our research hypothesis is that the training would improve the gait and balance by increasing the joint range of motion and extensor muscle activities in lower extremities and spatiotemporal measures.

## 2. Materials and Methods

### 2.1. Study Protocol and Participant Information

All study activities were conducted within the Center for Biomechanical & Rehabilitation Engineering (CBRE) at the University of the District of Columbia. The research study protocol was approved by the University of the District of Columbia IRB (#979744). In this paper, we show our results for a healthy control (male, 65 years old, Height 1.67 m, Mass: 84.5 kg) and a chronic stroke participant (female, 55 years old, left-side body impaired,

Height 1.54 m, Mass: 66.8 kg). Study participants also obtained the Mini-Mental State Examination (MMSE) score of 30 (perfect) indicating no cognitive impairment. Both participants were evaluated in the walking area under a multidirectional, overground robotic system (NaviGAITor) (Figure 1A). We structured our rehabilitation program to include redundant safety measures (spotters and NaviGAITor system). During training, routine inquiry was performed to check any training related discomfort and injuries.

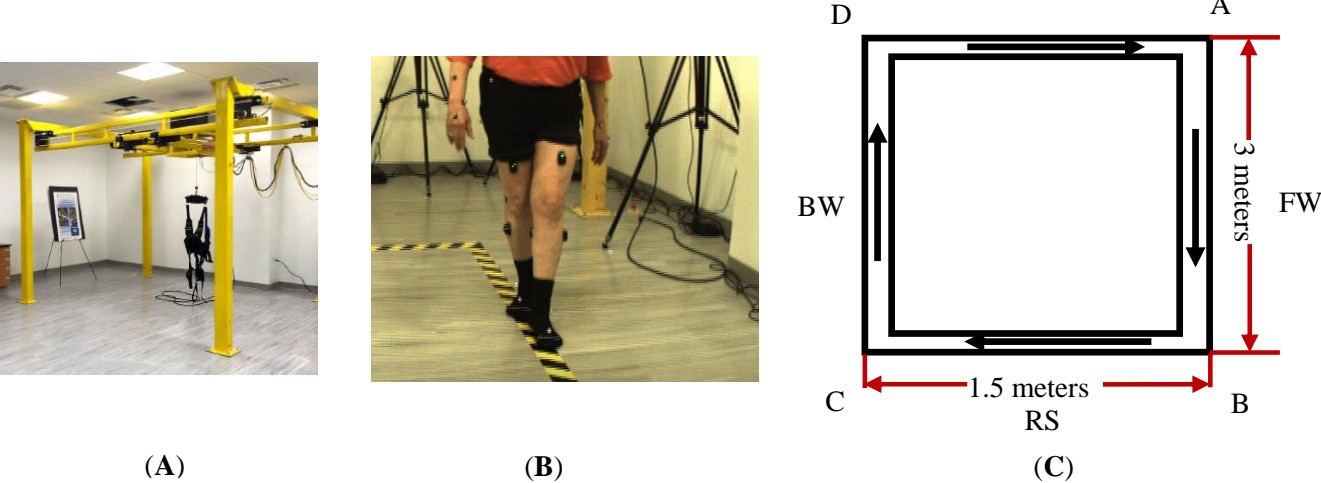

**Figure 1.** Gait training evaluation. (**A**) NaviGAITor; (**B**) participant performing tandem walk during FW section; (**C**) the dimensions of the assessment walking path. The arrows represent the walking directions.

### 2.2. Gait and Balance Assessment Protocol

Gait and balance were assessed before and after our sensorimotor training. Assessment was based on the joint kinematics, muscle activities and spatiotemporal data. The designated walking path is shown in Figure 1B,C. Participants started at point A, walked forward along AB, then moved parallel to BC while side stepping leading by the right leg, then performed backward walking for CD, and then finally, moved parallel to DA while side stepping leading by the left leg to get back to point A. For one rectangular cycle path, we therefore define these four segments in sequence as tandem forward walking (FW), right side stepping (RS), tandem backward walking (BW) and left side stepping (LS) as shown in Figure 1C. Participants were asked to perform tandem walking during the forward and backward walking sections. Tandem walking (heel to toe, or one foot in front of the other) is useful to screen patients for neurologic and vestibular disorders [16] as it can reveal inconsistencies and incongruencies tied to balance [17]. For walking and side stepping, we physically demonstrated how to do so along the paths marked by the black/yellow tape as shown in Figure 1B. All walking types used in the assessment were often used as part of mobility training to optimize walking ability in people with stroke [18]. The measurement protocol shown in Figure 2 summarizes the order of motions which were measured.

### 2.3. Multi-Sensorimotor Training

Participants completed a 6-week training routine that included two 30-min training sessions per week, 12 sessions in total. Participants wore neither shoes nor other footwear during both the training and assessment. The training was performed under conditions which targeted visual, vestibular and somatosensory sensory inputs. Visual inputs were manipulated by having eyes closed versus open whereas somatosensory inputs were manipulated by having participants stand on soft or medium density foam support versus a hard (floor) surface, or with wide versus narrow stances to modify the base of support. Vestibular inputs were targeted by having conditions wherein visual and/or somatosensory cues were limited (e.g., eyes closed while standing on foam or walking backwards on foam),

as well as spinning in a chair then having to stand and maintain balance. The training consisted of: Level (1), walking forwards and backwards eyes open or closed, regular and tandem along the path shown in Figure 1C; Level (2), foam-based standing or stepping and for varied stances eyes open or closed; Level (3), combined walking and foam-based exercise with increased sensory conflicts. More details about this training can be found in [19]. Each level consisted of 4 sessions. During each training session, NaviGAITor weight offloading system and spotters were present to prevent a fall.

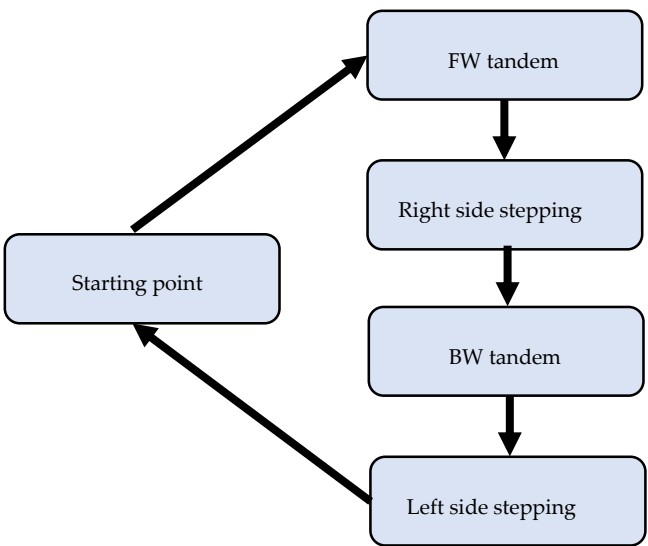

**Figure 2.** Flowchart of the motion and measurement protocol. Both the measurement and walking task orders were as follows: forward (FW) tandem, right side stepping, backward (BW) tandem and left side stepping.

### 2.4. Biomechanical Data Analysis

The data analysis pipeline is described below. Marker trajectories were recorded at 100 Hz from the Vicon MX system with 12 T40 cameras (Vicon Motion Systems, Oxford, UK). For joint angle analysis spatiotemporal measures, the placement of reflective markers follows the Vicon plugin gait Full-Body template (Figure 3) [20]. A total of 39 markers were attached to the anatomical landmarks of the body segments, 4 attached to the head (Left Forehead—LFHD, Right Forehead—RFHD, Left back of head—LBHD, Right back of head—RBHD), 14 attached to the upper limbs (7 to the right side, Right Shoulder—RSHO, Right Upper Arm—RUPA, Right Elbow—RELB, Right Forearm—RFRM, Right Wrist Marker A—RWRA, Right Wrist Marker B—RWRB, Right Finger—RFIN; 7 to the left side with the same arrangement), 5 attached to the torso (on the front: clavicle—CLAV, Sternum—STRN; on the back: 7th Cervical Vertebra—C7, Right Back—RBAK, 10th Thoracic Vertebra—T10), 4 attached to the pelvis (Left Anterior Superior iliac—LASI, Right Anterior Superior iliac—RASI, Left Posterior Superior iliac—LPSI, Right Posterior Superior iliac—RPSI), 12 attached to the lower limbs (6 to the right side, Right Thigh—RTHI, Right Knee—RKNE, Right Tibia—RTIB, Right Ankle—RANK, Right Heel—RHEE, Right Toe—RTOE; 6 to the left side with the same arrangement). The location of marker attachment is identified by palpating the corresponding anatomical landmarks.

Both static and dynamic motion trials of assessment are recorded and processed in Vicon Nexus software. Static trials from Vicon (Figure 3A) are loaded into Visual 3D (C-Motion, Gaithersburg, MD, USA) to reconstruct the human lower-extremity 3D model as shown in Figure 3B. Then, this static 3D model is applied to the tandem walking trial loaded from Vicon (shown in Figure 4A) into Visual 3D as shown in Figure 4C. Temporal data of marker trajectories from Vicon as shown in Figure 4B will be used to compute the lower-extremity joint angles in Visual 3D software as shown in Figure 4D. These joint angles

are flexion extension angles for the hip and the knee, and plantarflexion and dorsiflexion for the ankle.

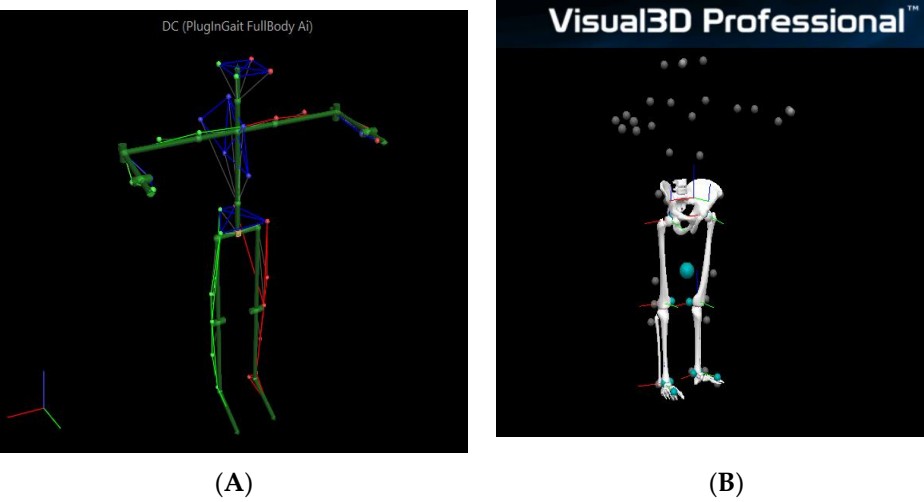

**(A)**                                                                 **(B)**

**Figure 3.** Illustration of model development for joint kinematics and spatiotemporal analysis. (**A**): view of marker-based participant profile during static trial using PluginGait Fullbody template in Nexus. The left bottom has the lab coordinates. The X axis is in a red color; Y axis is in a green color; Z axis is in a blue color. (**B**): Lower-limb Visual 3D Model-based maker data collecting from the static trial. The upper limb and torse are not modelled here in Visual 3D.

A velocity-based threshold at the foot center of gravity (COG) location was manually selected in Visual 3D software to identify the stance and swing phases of a gait cycle. Given that there were four walking segments, left and right legs, and pre- and post-evaluation conditions, we thus created 16 thresholds to cover all gait events of each participant. The gait cycle was based on identifying two types of gait events for one foot: heel strike (the start of weight loading) and toe off (the end of weight unloading). This identification method was suitable for the FW but not for the BW, LS, and RS. To better identify the stance and swing phases at three walking sections, we closely observed the targeted foot movement in Visual 3D and used the velocities of its center of gravity at the start of weight loading as the start of new stance phase and its velocities at the end of weight unloading as the start of new swing phase. During the side stepping, the step length and cadence of the trailing foot was reported as the same as the leading foot. The cadence of the trailing foot is calculated as a negative value during side stepping in Visual 3D. Therefore, we used the step length and cadence of the leading foot for both feet in LS and RS sections.

Muscle EMG activity was collected from four primary muscle groups (rectus femoris, semitendinosus also known as medial hamstring, tibialis anterior, medial gastrocnemius) from both legs by using a wireless EMG system (Trigno, Delsys, Boston, MA, USA) recorded at 2000 Hz. EMG data were band-pass filtered at 15 to 380 Hz, full-wave rectified, and low-pass filtered at 7 Hz to create a linear envelope [21]. Linear interpolation was used to normalize the data of both the joint angle and muscle activity. These data were then averaged across the gait cycles for each walking element. MATLAB (MathWorks, Natick, MA, USA) data processing scripts were created to generate all means and standard deviation plots of joint angles and muscle activities. Joint angles and EMG activities of the lower extremity and spatiotemporal parameters (cadence, step length, time used for the task) were used as outcome measures. The task completion time was calculated by finding the difference between the start and stop time of the whole walking task for each participant through observing the recorded task videos. The position change of trunk marker T10 in a primary direction aligned with the walking direction, which produced the average walking speed in each walking segment. These measurements in each walking segment were compared between pre and post training. The plots of joint angle and EMG activities for each walking segment of each leg show the mean value and standard deviation across

multiple gait cycles. The range of motion of each joint was calculated based on the difference between the maximum and minimum of the joint angles. The comparison of EMG activities between pre and post training was performed based on the averaged max EMG activities of each muscle at the selected phase (either stance or swing). A two-sample t-test (ttest2 in MATLAB) was performed to study the change of two outcome measurements (joint angle and max muscle activities) from pre to post for each participant. The joint ROM and max muscle activity at the post is considered significantly different from those at the pre training if the *p* value is less than 0.05. The overall data-processing protocol is summarized in Figure 5.

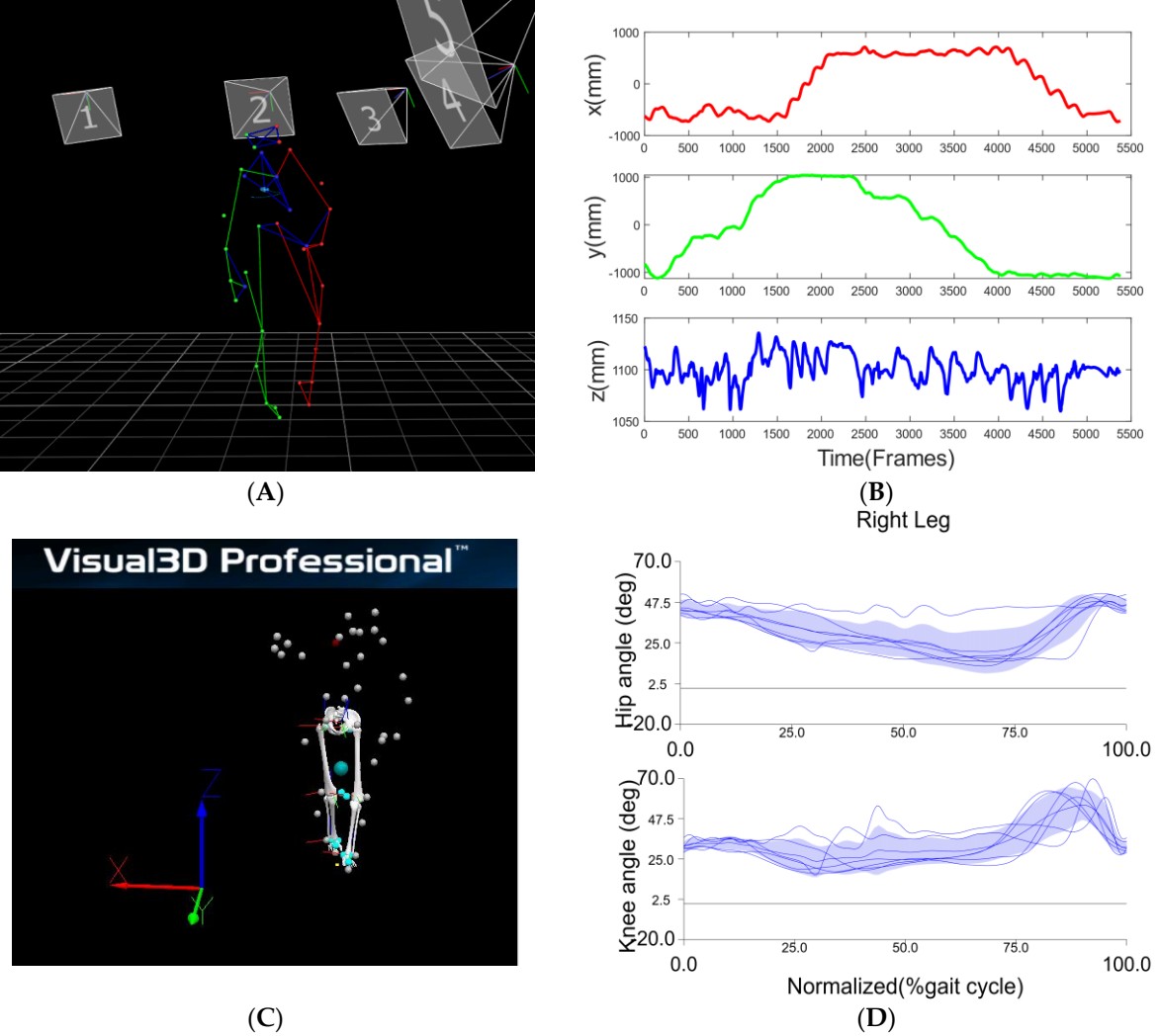

**Figure 4.** Illustration of data processing with time-course data of markers. (**A**) 3D perspective view of marker-based participant profile during the tandem forward walking in Nexus; (**B**) An example of X, Y, Z positions of the marker T10 with respect to the lab coordinates along the recorded trial time, each of which is presented by the blue line. (**C**) Visual 3D modelling of marker-based participant profile at the same time frame; the upper limb and torso are not modelled here in Visual 3D. (**D**) An example of right hip and knee angles in forward walking section computed from marker trajectories using inverse kinematics in Visual 3D software. The thin blue lines represent the joint angle of each gait cycle. The blue shaded regions represent the mean and standard deviation of the joint angle across all gait cycles.

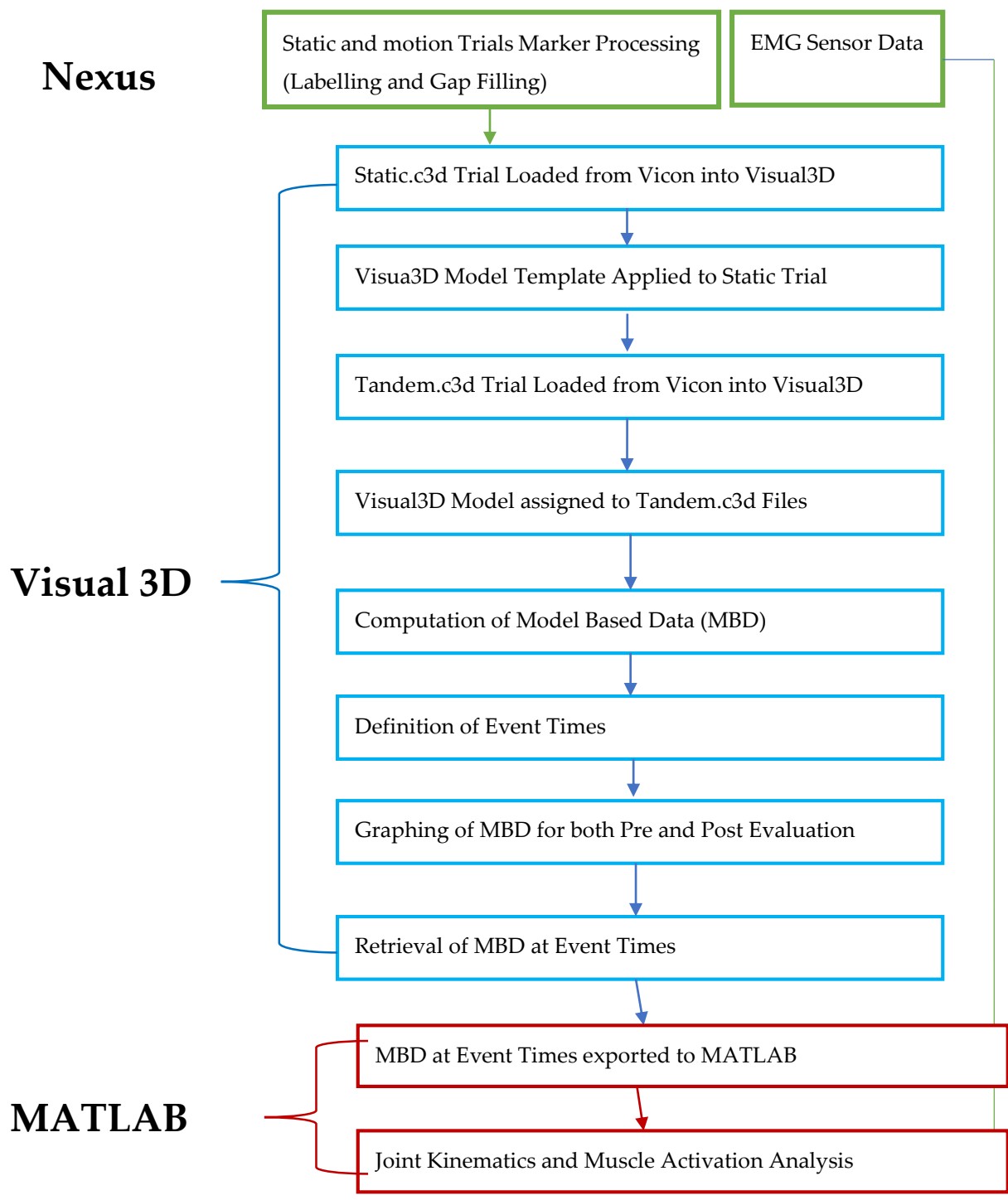

**Figure 5.** Flowchart of data processing protocol. A static.c3d represents a static pose trial and a tandem.c3d represents a motion trial.

### 3. Results

*3.1. Healthy Participant Kinematics and EMG Analysis*

The recorded video in the tandem condition showed that the healthy participant used 27 s to complete the walking path during the post-assessment, compared to 24 s during the preassessment. This slightly decreased speed for walking completion can be seen in the decreased average walking speed at each walking segment during post evaluation, shown

in Table 1. The decreased step lengths of both legs in most walking segments (as shown in Table 2) contribute to the reduced walking speed.

**Table 1.** Comparison of average walking speed at each walking segment of healthy participant.

| Walking Speed (m/s) | FW | RS | BW | LS |
|---|---|---|---|---|
| pre | 0.39 | 0.27 | 0.26 | 0.29 |
| post | 0.34 | 0.22 | 0.25 | 0.20 |

Note: FW: forward walking; RS: the right sidestep; BW: backward walking; LS: the left sidestep.

**Table 2.** Primary Spatio-Temporal Parameters from the gait assessment for control participant.

| Parameters | Pre Training (Baseline) | | Post Training | |
|---|---|---|---|---|
| Left/Right (L/R) | L | R | L | R |
| No. of Gait Cycles (FW) | 2 | 3 | 2 | 2 |
| Step Length (FW) (m) | 0.371 (0.070) | 0.338 (0.037) | 0.300 (0.115) | 0.316 (0.044) |
| Cadence (FW) (steps/min) | 69 (4) | 78 (18) | 75 (7) | 68 (1) |
| No. of Gait Cycles (RS) | 3 | 4 | 5 | 4 |
| Step Length (RS) (m) | 0.40 (0.073) | 0.40 (0.073) | 0.337 (0.071) | 0.337 (0.071) |
| Cadence (RS) (steps/min) | 85 (11) | 85 (11) | 77 (11) | 77 (11) |
| No. of Gait Cycles (BW) | 4 | 4 | 4 | 4 |
| Step Length (BW) (m) | 0.184 (0.043) | 0.249 (0.150) | 0.229 (0.094) | 0.171 (0.135) |
| Cadence (BW) (steps/min) | 68 (9) | 64 (4) | 54 (10) | 93 (20) |
| No. of Gait Cycles (LS) | 4 | 3 | 3 | 4 |
| Step Length (LS) (m) | 0.417 (0.014) | 0.417 (0.014) | 0.388 (0.014) | 0.388 (0.014) |
| Cadence (LS) (steps/min) | 86 (2) | 86 (2) | 80 (6) | 80 (6) |

Note: FW: forward walking; RS: the right sidestep; BW: backward walking; LS: the left sidestep. However, the healthy participant walked the training path in the opposite direction during his pre training. Therefore, his left foot was the leading foot in his RS section, whereas his right foot was the leading foot in his LS section. The step length and cadence of the trailing feet were reported same as the leading feet in RS and LS sections for both pre and past training. The values shown are mean (SD).

The joint angle profiles are displayed in Figure 6. The min, max and range of motion values for each joint angle are shown in Table 3. Compared to pre assessment, there is a joint angle profile shift happening to most lower extremity joints on both legs in all four walking segments. However, most joint range of motions (ROM) remain statistically same; but there were a few exceptions. The ROMs of hip and knee decreased significantly by 3.7 degrees ($p = 0.015$) and 4.2 degrees ($p = 0.007$) on the left leg during the right-side stepping. During the backward walking, the ROM of the left ankle joint decreased at a marginal significant level by 5.1 degrees ($p = 0.048$). During the left-side stepping, the ROM of the right knee joint increased significantly by 4.2 degrees ($p = 0.031$); the ROM of the left ankle joint increased significantly by 5.5 degrees ($p = 0.006$) whereas the ROM of the right ankle decreased significantly by 2.9 degrees ($p = 0.009$).

The muscle activity profiles are displayed in Figure 7. The max of each muscle activity is shown in Table 4. When performing forward walking, the activity of the tibial anterior on the left leg ($p = 0.038$) and rectus femoris on the right leg ($p = 0.017$) increased significantly whereas the activity of semitendinosus significantly decreased on the right leg ($p = 0.005$) during the swing phase, as shown in Figure 7A.

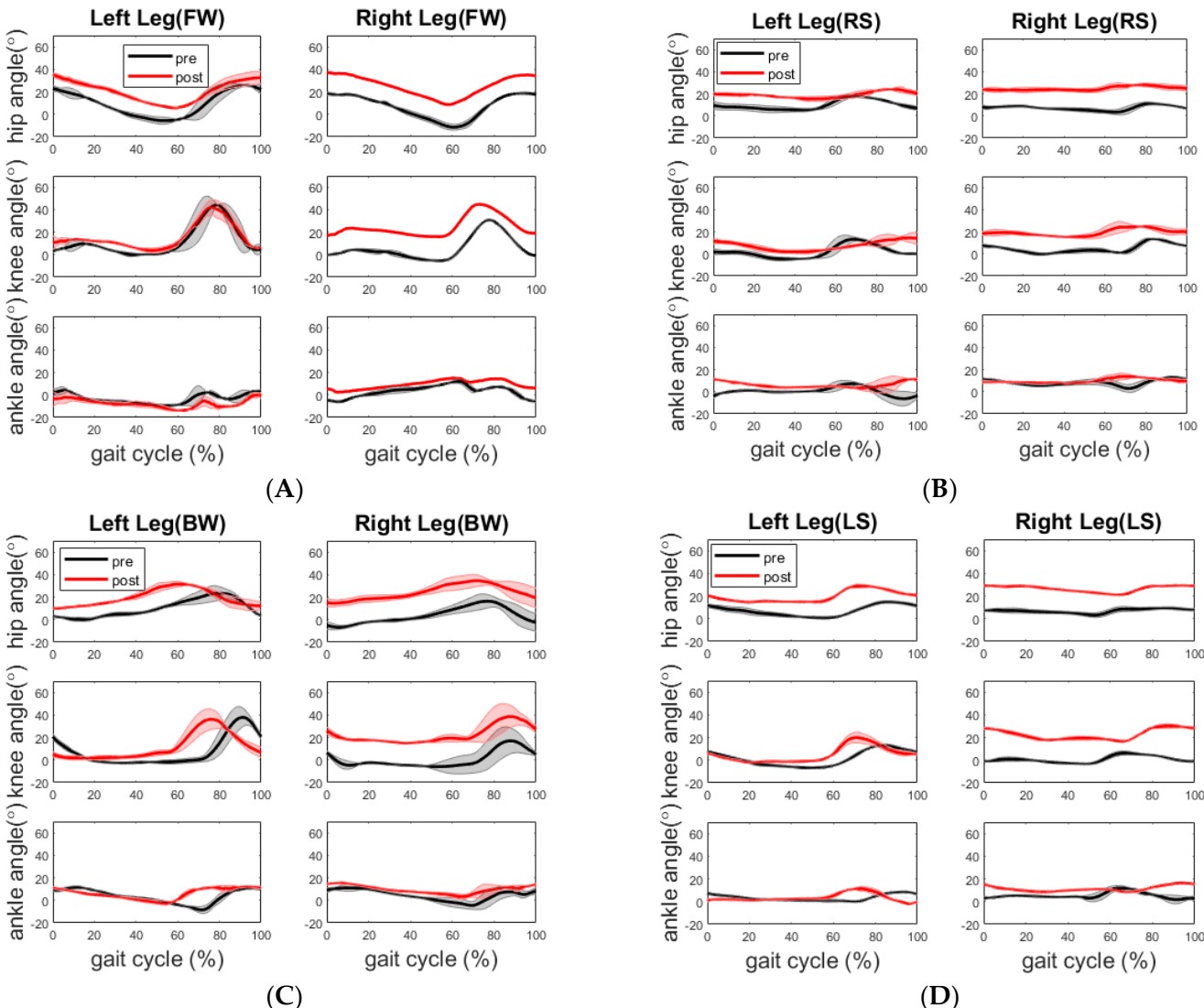

**Figure 6.** Comparison of mean joint angles across the gait cycle during tandem walking condition before and after our training for a healthy participant. (**A**) Forward walking; (**B**) Right sidestep; (**C**) Backward walking; (**D**) Left sidestep. All plots are mean values with standard deviation as the shaded region.

For extensor muscles performing the right-side stepping (shown in Figure 7B), the activity of the tibial anterior of the left leg increased significantly during the stance phase ($p = 0.007$) whereas the activity of the same muscle group on the right leg decreased significantly ($p = 0.020$) during the swing phase. Rectus femoris muscle activity increased marginally during the swing phase ($p = 0.051$) on the right side.

For extensor muscles performing the backward walking (shown in Figure 7C), the activity of the tibial anterior of both legs decreased significantly during the stance phase (left: $p = 0.003$; right: $p = 0.0005$). The rectus femoris muscle activity of both legs increased significantly during the stance phase. For the left leg, $p = 0.012$. For the right leg, $p < 0.001$. For flexor muscles performing the backward walking, the activity of the semitendinosus on the left leg decreased significantly during the stance phase ($p < 0.001$).

**Table 3.** Comparison of joint angle min max and range of motion for healthy participant.

| Joint Angle | FW | Min (deg) | | Max (deg) | | ROM (deg) | | ROM Diff. | *p* Value |
| | | Pre | Post | Pre | Post | Pre | Post | | |
|---|---|---|---|---|---|---|---|---|---|
| Hip–FW | L | −6.1 ± 2.5 | 5.5 ± 0.5 | 26.1 ± 0.4 | 36.2 ± 0.3 | 32.2 ± 2.8 | 30.7 ± 0.2 | −1.5 ± 2.6 | 0.524 |
| | R | −11.5 ± 2.6 | 8.6 ± 0.0 | 18.7 ± 0.5 | 37.3 ± 0.0 | 30.2 ± 3.1 | 28.7 ± 0.0 | −1.5 ± 3.1 | 0.747 |
| Knee–FW | L | −0.3 ± 1.5 | 3.7 ± 2.0 | 46.5 ± 2.6 | 42.3 ± 6.2 | 46.8 ± 4.2 | 38.6 ± 4.2 | −8.2 ± 8.4 | 0.191 |
| | R | −5.5 ± 0.3 | 16.0 ± 0.0 | 30.7 ± 1.1 | 44.7 ± 0.0 | 36.2 ± 0.8 | 28.7 ± 0.0 | −7.5 ± 0.8 | 0.084 |
| Ankle–FW | L | −9.6 ± 0.0 | −13.9 ± 0.6 | 5.3 ± 1.4 | 1.7 ± 0.1 | 14.9 ± 1.4 | 15.6 ± 0.5 | 0.7 ± 1.9 | 0.572 |
| | R | −6.2 ± 0.6 | 2.4 ± 0.0 | 12.2 ± 1.9 | 15.0 ± 0.0 | 18.4 ± 2.5 | 12.6 ± 0.0 | −5.8 ± 2.5 | 0.305 |
| Hip–RS | L | 4.6 ± 1.6 | 15.0 ± 2.3 | 18.2 ± 1.0 | 24.9 ± 2.0 | 13.6 ± 0.7 | 9.9 ± 3.8 | −3.7 ± 1.7 | 0.015 |
| | R | 3.1 ± 2.2 | 22.3 ± 1.8 | 11.3 ± 1.3 | 29.0 ± 1.1 | 8.2 ± 2.2 | 6.7 ± 2.7 | −1.5 ± 3.9 | 0.174 |
| Knee–RS | L | −4.9 ± 1.4 | 1.5 ± 1.6 | 14.3 ± 2.4 | 16.7 ± 3.2 | 19.4 ± 1.7 | 15.2 ± 4.8 | −4.2 ± 2.9 | 0.007 |
| | R | −0.6 ± 1.0 | 14.8 ± 1.3 | 13.6 ± 0.9 | 26.5 ± 0.5 | 14.2 ± 1.8 | 11.7 ± 0.9 | −2.5 ± 1.2 | 0.063 |
| Ankle–RS | L | −8.2 ± 3.5 | 1.2 ± 3.9 | 7.8 ± 2.1 | 12.0 ± 0.5 | 16.0 ± 5.6 | 10.8 ± 3.9 | −5.2 ± 4.8 | 0.093 |
| | R | 1.7 ± 2.5 | 7.4 ± 0.3 | 13.2 ± 0.3 | 14.8 ± 1.1 | 11.5 ± 2.4 | 7.4 ± 1.0 | −4.1 ± 2.2 | 0.069 |
| Hip–BW | L | −0.0 ± 1.7 | 9.6 ± 0.3 | 24.4 ± 5.2 | 32.0 ± 1.8 | 24.4 ± 6.7 | 22.3 ± 2.0 | −2.1 ± 4.7 | 0.627 |
| | R | −7.9 ± 1.6 | 14.0 ± 3.1 | 19.1 ± 4.2 | 36.8 ± 2.4 | 27.0 ± 5.3 | 22.8 ± 5.1 | −4.2 ± 1.2 | 0.307 |
| Knee–BW | L | −3.2 ± 0.7 | 1.3 ± 1.4 | 38.8 ± 7.5 | 37.1 ± 8.4 | 42.0 ± 7.0 | 35.9 ± 7.0 | −6.1 ± 0.7 | 0.339 |
| | R | −9.0 ± 4.8 | 14.7 ± 0.6 | 20.2 ± 9.6 | 42.1 ± 8.2 | 29.2 ± 9.5 | 27.4 ± 7.9 | −1.8 ± 6.5 | 0.769 |
| Ankle–BW | L | −8.7 ± 3.3 | −2.5 ± 1.8 | 11.7 ± 1.4 | 12.8 ± 0.6 | 20.4 ± 2.8 | 15.3 ± 1.5 | −5.1 ± 1.3 | 0.048 |
| | R | −5.1 ± 3.2 | 2.2 ± 2.5 | 12.5 ± 1.0 | 15.6 ± 0.9 | 17.6 ± 3.6 | 13.4 ± 3.5 | −4.2 ± 3.3 | 0.142 |
| Hip–LS | L | 0.8 ± 0.2 | 14.1 ± 0.7 | 14.8 ± 1.3 | 28.9 ± 1.5 | 14.0 ± 1.4 | 14.8 ± 2.0 | 0.8 ± 3.1 | 0.623 |
| | R | 3.0 ± 2 | 21.1 ± 1.8 | 10.2 ± 0.9 | 29.9 ± 1.1 | 7.2 ± 1.1 | 8.8 ± 2.7 | 1.6 ± 2.2 | 0.164 |
| Knee–LS | L | −6.8 ± 0.4 | −2.2 ± 0.3 | 13.0 ± 1.3 | 20.2 ± 4.5 | 19.8 ± 1.4 | 22.4 ± 4.8 | 2.6 ± 3.9 | 0.411 |
| | R | −3.3 ± 0.9 | 16.6 ± 1.3 | 6.3 ± 1.8 | 30.4 ± 0.5 | 9.6 ± 1.4 | 13.8 ± 0.9 | 4.2 ± 3.0 | 0.031 |
| Ankle–LS | L | 0.0 ± 0.9 | −2.1 ± 0.6 | 8.5 ± 0.7 | 11.9 ± 1.6 | 8.5 ± 1.5 | 14.0 ± 1.0 | 5.5 ± 0.4 | 0.006 |
| | R | 0.7 ± 2.8 | 8.0 ± 0.3 | 12.1 ± 1.8 | 16.5 ± 1.1 | 11.4 ± 1.1 | 8.5 ± 1.0 | −2.9 ± 1.2 | 0.009 |

Note: FW: forward walking segment. RS: right side-stepping segment. BW: backward walking segment. LS: left side-stepping segment. ST: Stance phase; SW: swing phase. The values shown are mean (SD).

**Table 4.** Comparison of leg muscle activities during stance and swing phase for healthy participant.

| Gait Phase | Side | Medial Gastrocnemius (µV) | | | Semitendinosus (µV) | | | Tibial Anterior (µV) | | | Rectus Femoris (µV) | | |
| | | Pre | Post | *p* Value | Pre | Post | *p* Value | Pre | Post | *p* Value | Pre | Post | *p* Value |
|---|---|---|---|---|---|---|---|---|---|---|---|---|---|
| FW–ST | L | 77 ± 62 | 116 ± 84 | 0.651 | 58 ± 44 | 39 ± 10 | 0.600 | 69 ± 42 | 36 ± 1 | 0.383 | 28 ± 6 | 57 ± 21 | 0.204 |
| | R | 43 ± 22 | 31 ± 7 | 0.530 | 155 ± 48 | 110 ± 52 | 0.464 | 49 ± 17 | 24 ± 5 | 0.193 | 17 ± 7 | 37 ± 5 | 0.077 |
| RS–ST | L | 21 ± 5 | 36 ± 9 | 0.075 | 37 ± 18 | 21 ± 2 | 0.199 | 22 ± 2 | 43 ± 7 | 0.007 | 21 ± 8 | 33 ± 20 | 0.396 |
| | R | 39 ± 14 | 17 ± 6 | 0.103 | 124 ± 39 | 85 ± 35 | 0.247 | 32 ± 9 | 27 ± 8 | 0.454 | 37 ± 12 | 36 ± 19 | 0.951 |
| BW–ST | L | 32 ± 12 | 24 ± 9 | 0.370 | 25 ± 2 | 12 ± 3 | <0.001 | 69 ± 8 | 42 ± 5 | 0.012 | 31 ± 7 | 52 ± 7 | 0.014 |
| | R | 68 ± 30 | 34 ± 13 | 0.086 | 96 ± 65 | 57 ± 22 | 0.300 | 62 ± 13 | 18 ± 2 | <0.001 | 22 ± 2 | 19 ± 7 | 0.457 |
| LS–ST | L | 54 ± 4 | 15 ± 2 | <0.001 | 48 ± 9 | 12 ± 4 | 0.003 | 56 ± 12 | 36 ± 9 | 0.0872 | 35 ± 14 | 33 ± 9 | 0.854 |
| | R | 27 ± 4 | 18 ± 8 | 0.166 | 244 ± 147 | 63 ± 24 | 0.102 | 49 ± 18 | 20 ± 2 | 0.051 | 29 ± 9 | 14 ± 0 | 0.044 |
| FW–SW | L | 16 ± 3 | 11 ± 2 | 0.240 | 47 ± 27 | 8 ± 0 | 0.178 | 16 ± 2 | 23 ± 1 | 0.038 | 15 ± 1 | 60 ± 17 | 0.062 |
| | R | 31 ± 24 | 5 ± 2 | 0.272 | 392 ± 31 | 82 ± 6 | 0.005 | 22 ± 1 | 33 ± 19 | 0.516 | 10 ± 5 | 34 ± 1 | 0.017 |
| RS–SW | L | 9 ± 1 | 12 ± 11 | 0.620 | 22 ± 8 | 18 ± 5 | 0.492 | 21 ± 3 | 26 ± 3 | 0.084 | 13 ± 2 | 40 ± 26 | 0.157 |
| | R | 22 ± 8 | 10 ± 6 | 0.094 | 55 ± 8 | 45 ± 12 | 0.2519 | 31 ± 4 | 17 ± 7 | 0.020 | 7 ± 1 | 30 ± 15 | 0.051 |
| BW–SW | L | 28 ± 24 | 48 ± 28 | 0.354 | 36 ± 10 | 17 ± 3 | 0.072 | 21 ± 3 | 33 ± 10 | 0.076 | 15 ± 2 | 18 ± 4 | 0.252 |
| | R | 19 ± 8 | 36 ± 29 | 0.311 | 66 ± 19 | 43 ± 24 | 0.185 | 33 ± 9 | 52 ± 29 | 0.253 | 13 ± 5 | 12 ± 1 | 0.662 |
| LS–SW | L | 19 ± 14 | 5 ± 2 | 0.163 | 16 ± 4 | 10 ± 1 | 0.067 | 18 ± 4 | 22 ± 4 | 0.334 | 13 ± 1 | 40 ± 16 | 0.043 |
| | R | 16 ± 3 | 10 ± 6 | 0.170 | 51 ± 9 | 57 ± 16 | 0.610 | 37 ± 3 | 12 ± 4 | <0.001 | 9 ± 1 | 18 ± 4 | 0.020 |

Note: FW: forward walking segment. RS: right side-stepping segment. BW: backward walking segment. LS: left side-stepping segment. ST: Stance phase; SW: swing phase. The values shown are mean (SD).

As shown in Figure 7D, for extensor muscles performing the left sidestep walking, the activity of the tibial anterior on the right leg decreased marginally during the stance phase ($p = 0.051$) and decreased significantly during the swing phase ($p < 0.001$). The activity of the rectus femoris decreased significantly on its right leg during the stance phase ($p = 0.044$) whereas this muscle activity increased significantly on both legs during the swing phase (left: $p = 0.043$; right: $p = 0.020$). For flexor muscles performing this sidestep, the medial

gastrocnemius muscle activity on the left leg decreased significantly ($p = 0.003$) during the stance phase. The semitendinosus muscle activity on the left leg decreased significantly during the stance phase ($p = 0.003$).

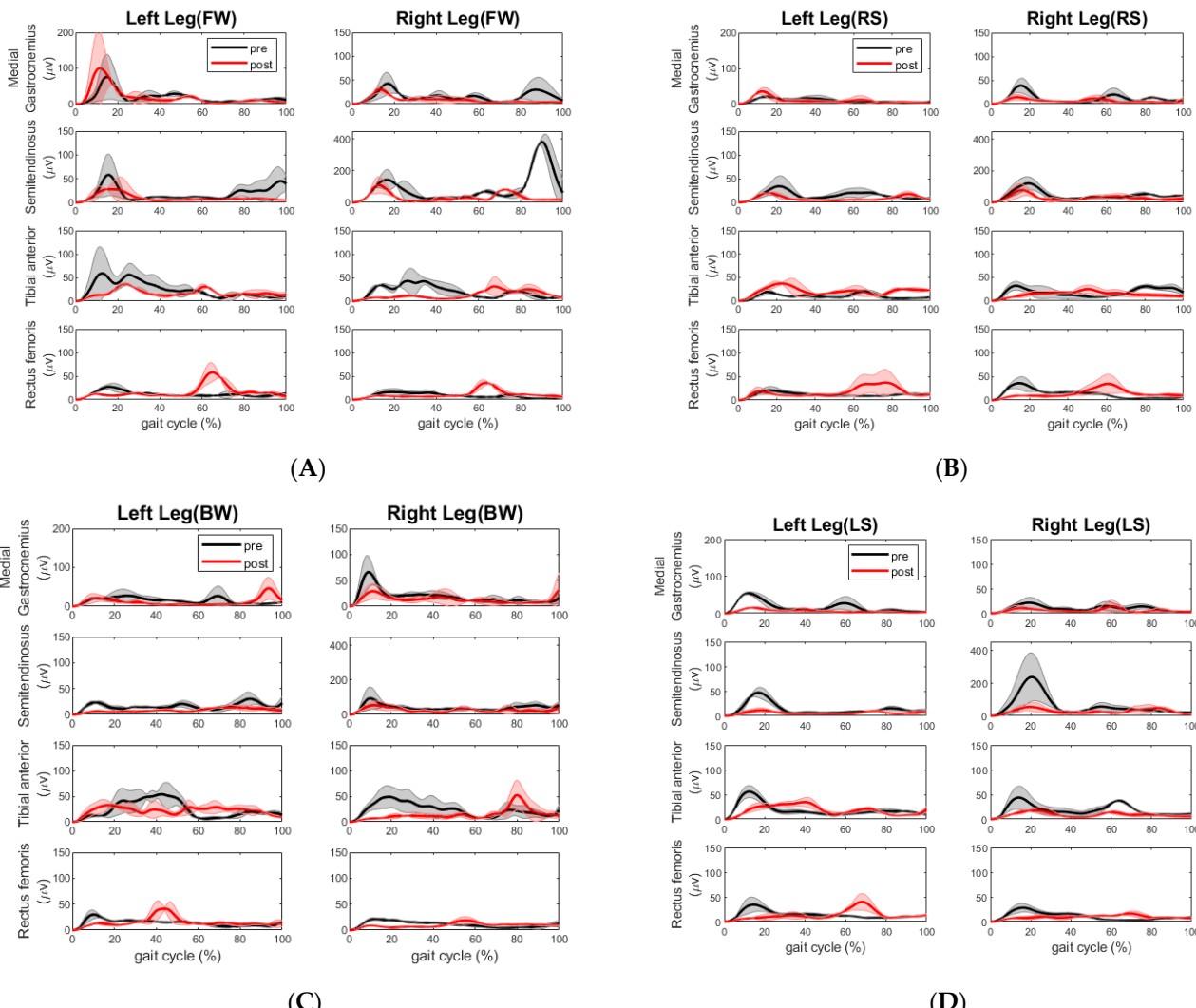

**Figure 7.** Mean EMG linear envelopes (in muV) across the gait cycle during the pre (black) and post (red) training tandem walking conditions for the control participant. The shaded regions represent ±1 SD from the mean in each walking element. (**A**) Forward walking; (**B**) Right sidestep; (**C**) Backward walking; (**D**) Left sidestep. All plots are mean values with standard deviation as the shaded region.

### 3.2. Stroke Participant Kinematics and EMG Analysis

The recorded videos show that in the tandem condition, this participant used 23 s to complete the walking path during the post assessment, compared to 42 s during the pre-assessment. The reduced time for task completion can be seen in the increased average walking speed at each walking segment during post evaluation shown in Table 5. The increased step lengths of both legs in most walking segments (as shown in Table 6) contribute to the increased walking speed. Compared to pre assessment, gait smoothness improved due to increased step length in the left impaired leg.

**Table 5.** Comparison of average walking speed at each walking segment of stroke participant.

| Walking Speed (m/s) | FW | RS | BW | LS |
|---|---|---|---|---|
| pre | 0.15 | 0.23 | 0.11 | 0.14 |
| post | 0.29 | 0.27 | 0.23 | 0.19 |

**Table 6.** Primary Spatiotemporal Parameters from the gait assessment for stroke participant.

| Parameters | Pre Training (Baseline) | | Post Training | |
|---|---|---|---|---|
| Left/Right (L/R) | L | R | L | R |
| No. of Gait Cycles (FW) | 4 | 4 | 3 | 3 |
| Step Length (FW) (m) | 0.184 (0.081) | 0.257 (0.091) | 0.381 (0.041) | 0.237 (0.025) |
| Candence (FW) (steps/min) | 48 (8) | 67 (17) | 58 (7) | 82 (5) |
| No. of Gait Cycles (RS) | 3 | 3 | 3 | 2 |
| Step Length (RS) (m) | 0.367 (0.054) | 0.367 (0.054) | 0.469 (0.032) | 0.469 (0.032) |
| Candence (FW) (steps/min) | 94 (6) | 94 (6) | 97 (7) | 97 (7) |
| No. of Gait Cycles (BW) | 5 | 5 | 4 | 4 |
| Step Length (BW) (m) | 0.250 (0.060) | 0.141 (0.059) | 0.285 (0.053) | 0.228 (0.058) |
| Candence (FW) (steps/min) | 42 (18) | 44 (13) | 54 (8) | 71 (16) |
| No. of Gait Cycles (LS) | 5 | 5 | 3 | 4 |
| Step Length (LS) (m) | 0.366 (0.044) | 0.366 (0.044) | 0.376 (0.063) | 0.376 (0.063) |
| Candence (FW) (steps/min) | 91 (35) | 91 (35) | 83 (3) | 83 (3) |

Note: The step length and cadence of the trailing feet were reported the same as the leading feet in RS and LS sections. The values shown are mean (SD).

In general, the joint range of motion did not change for most walking segments as shown in Figure 8 and Table 7, but there are a few exceptions. During the backward walking, the ROM of the hip joint increased significantly by 5.8 degrees ($p = 0.019$) on the left leg side, whereas the ROMs of the hip joint and knee joint increased significantly by 4.1 degrees ($p = 0.046$) and 8.1 degrees ($p = 0.007$) on the right leg side.

Most muscle activities did not change for all walking segments as shown in Figure 9 and Table 8, but there are a few exceptions. When performing the right-side stepping, rectus femoris muscle activity decreased marginally during both the stance phase ($p = 0.052$) and the swing phase ($p = 0.049$) on the left leg; the semitendinosus muscle activity decreased significantly during the swing phase ($p = 0.004$). As shown in Figure 9C, when performing the backward walking, the activity of the extensor rectus femoris on the right leg increased significantly during the stance phase ($p = 0.005$) and during the swing phase as well ($p = 0.014$), the activity of the flexor medial gastrocnemius on the right leg decreased significantly during the stance phase ($p = 0.002$), and the activity of the semitendinosus increased significantly during the swing phase ($p = 0.044$). As shown in Figure 9D, when performing the left side stepping, the activity of the extensor medial gastrocnemius decreased significantly on the right leg during the stance phase ($p = 0.018$).

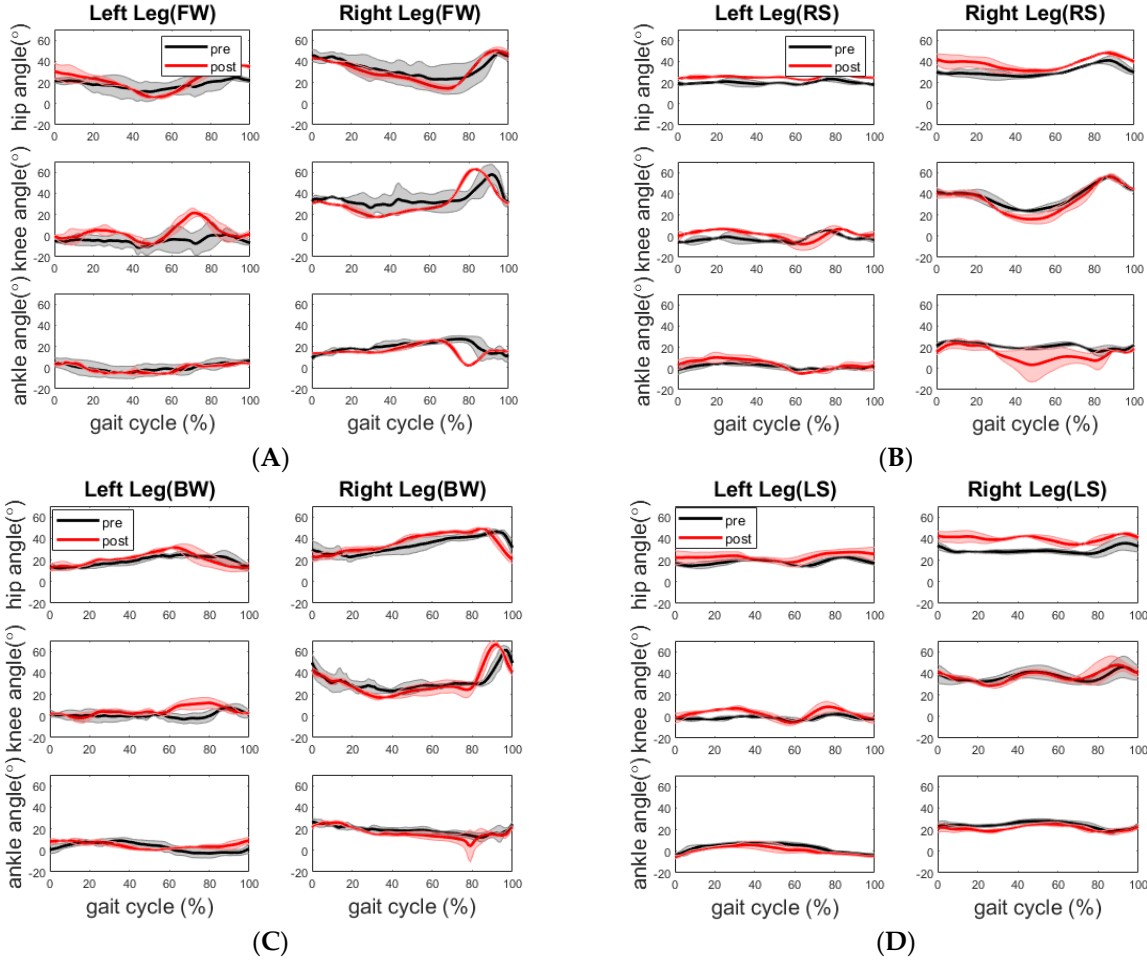

**Figure 8.** Comparison of mean joint angles across the gait cycle during tandem walking condition before and after training for the stroke participant. (**A**) Forward walking; (**B**) Right sidestep; (**C**) Backward walking; (**D**) Left sidestep. All plots are mean values with standard deviation as the shaded region.

**Table 7.** Comparison of joint angle min/max and range of motion for stroke participant.

| Joint Angle | FW | Min (deg) | | Max (deg) | | ROM (deg) | | ROM Diff. | *p* Value |
|---|---|---|---|---|---|---|---|---|---|
| | | Pre | Post | Pre | Post | Pre | Post | | |
| Hip–FW | L | 7.0 ± 8.5 | 5.6 ± 0.7 | 26.9 ± 3.2 | 37.0 ± 2.1 | 19.9 ± 11.4 | 31.4 ± 2.4 | 11.5 ± 16.6 | 0.085 |
| | R | 20.9 ± 11.4 | 14.4 ± 3.0 | 51.0 ± 1.0 | 50.2 ± 3.1 | 30.1 ± 10.9 | 35.7 ± 0.1 | 14.6 ± 14.3 | 0.377 |
| Knee–FW | L | −16.5 ± 8.5 | −9.8 ± 1.3 | 10.0 ± 8.8 | 22.6 ± 2.3 | 26.5 ± 14.7 | 32.4 ± 3.5 | 5.9 ± 18.2 | 0.428 |
| | R | 20.8 ± 5.4 | 17.4 ± 0.7 | 64.1 ± 5.4 | 62.8 ± 0.5 | 43.3 ± 9.8 | 45.4 ± 1.1 | 2.1 ± 11.3 | 0.704 |
| Ankle–FW | L | −5.5 ± 6.1 | −7.0 ± 0.1 | 8.5 ± 2.8 | 5.7 ± 0.8 | 14.0 ± 5.8 | 12.7 ± 0.6 | −1.3 ± 0.65 | 0.636 |
| | R | 5.1 ± 1.4 | 2.0 ± 0.4 | 27.5 ± 2.7 | 26.0 ± 0.6 | 22.4 ± 3.6 | 24.0 ± 1.0 | 1.6 ± 2.0 | 0.488 |
| Hip–RS | L | 17.3 ± 0.7 | 22.5 ± 1.0 | 23.5 ± 2.5 | 28.9 ± 0.6 | 6.2 ± 1.8 | 6.4 ± 0.4 | 0.2 ± 2.2 | 0.880 |
| | R | 25.5 ± 2.4 | 30.9 ± 1.9 | 41.4 ± 3.4 | 48.1 ± 1.4 | 15.9 ± 3.7 | 17.2 ± 3.3 | 1.3 ± 6.2 | 0.700 |
| Knee–RS | L | −7.5 ± 0.5 | −7.5 ± 5.7 | 5.4 ± 1.0 | 7.2 ± 2.0 | 12.9 ± 0.5 | 14.7 ± 3.7 | 1.8 ± 3.2 | 0.583 |
| | R | 23.5 ± 1.1 | 15.9 ± 4.1 | 56.4 ± 1.8 | 56.9 ± 1.6 | 32.7 ± 2.3 | 41.0 ± 5.7 | 8.3 ± 8.9 | 0.263 |
| Ankle–RS | L | −3.2 ± 1.5 | −4.7 ± 0.2 | 6.1 ± 1.2 | 10.6 ± 4.6 | 9.3 ± 2.7 | 15.3 ± 4.5 | 6.0 ± 7.1 | 0.242 |
| | R | 15.3 ± 3.4 | 2.2 ± 14.4 | 26.1 ± 0.6 | 23.6 ± 4.7 | 10.8 ± 3.3 | 21.4 ± 19.1 | 10.6 ± 23.6 | 0.578 |
| Hip–BW | L | 11.4 ± 0.9 | 10.8 ± 3.2 | 27.3 ± 2.5 | 32.5 ± 1.6 | 15.9 ± 2.8 | 21.7 ± 1.6 | 5.8 ± 2.4 | 0.019 |
| | R | 20.7 ± 2.8 | 18.8 ± 1.4 | 47.4 ± 2.4 | 49.6 ± 0.9 | 26.7 ± 2.9 | 30.8 ± 1.5 | 4.1 ± 3.7 | 0.046 |

**Table 7.** *Cont.*

| Joint Angle | FW | Min (deg) | | Max (deg) | | ROM (deg) | | ROM Diff. | *p* Value |
|---|---|---|---|---|---|---|---|---|---|
| | | Pre | Post | Pre | Post | Pre | Post | | |
| Knee–BW | L | $-6.4 \pm 2.7$ | $-2.2 \pm 4.3$ | $9.5 \pm 3.2$ | $12.6 \pm 5.1$ | $15.9 \pm 5.0$ | $14.8 \pm 9.2$ | $-1.1 \pm 8.7$ | 0.8425 |
| | R | $20.3 \pm 1.7$ | $15.8 \pm 1.1$ | $63.4 \pm 3.1$ | $67.0 \pm 2.3$ | $43.1 \pm 2.5$ | $51.2 \pm 3.1$ | $8.1 \pm 2.9$ | 0.007 |
| Ankle–BW | L | $-4.3 \pm 4.0$ | $-0.3 \pm 0.6$ | $10.8 \pm 1.5$ | $10.8 \pm 0.6$ | $15.1 \pm 4.4$ | $11.1 \pm 1.1$ | $-6.6 \pm 3.9$ | 0.180 |
| | R | $8.6 \pm 4.1$ | $1.7 \pm 12.4$ | $27.8 \pm 1.8$ | $26.3 \pm 1.8$ | $19.1 \pm 4.6$ | $24.6 \pm 10.9$ | $5.5 \pm 14.6$ | 0.391 |
| Hip–LS | L | $13.3 \pm 0.9$ | $17.0 \pm 3.3$ | $23.3 \pm 1.1$ | $28.4 \pm 3.2$ | $10 \pm 2.0$ | $11.4 \pm 1.1$ | $1.4 \pm 2.3$ | 0.280 |
| | R | $24.6 \pm 1.2$ | $33.4 \pm 2.4$ | $38.3 \pm 2.3$ | $46.2 \pm 1.7$ | $13.7 \pm 2.1$ | $12.8 \pm 4.0$ | $-0.9 \pm 4.4$ | 0.735 |
| Knee–LS | L | $-5.4 \pm 1.4$ | $-6.6 \pm 2.0$ | $2.9 \pm 1.8$ | $10.7 \pm 3.3$ | $8.3 \pm 2.1$ | $17.3 \pm 4.6$ | $9.0 \pm 6.7$ | 0.062 |
| | R | $28.5 \pm 3.4$ | $28.1 \pm 1.8$ | $49.1 \pm 3.4$ | $48.6 \pm 6.9$ | $20.6 \pm 3.9$ | $20.5 \pm 6.8$ | $-0.1 \pm 4.1$ | 0.990 |
| Ankle–LS | L | $-3.9 \pm 1.3$ | $-5.8 \pm 1.4$ | $8.6 \pm 0.7$ | $6.2 \pm 2.7$ | $12.5 \pm 1.6$ | $12.0 \pm 3.9$ | $-0.5 \pm 5.2$ | 0.852 |
| | R | $17.4 \pm 2.3$ | $16.0 \pm 1.1$ | $28.5 \pm 1.4$ | $25.5 \pm 1.9$ | $11.1 \pm 3.5$ | $9.5 \pm 1.1$ | $-2.6 \pm 1.8$ | 0.393 |

Note: FW: forward walking segment. RS: right side-stepping segment. BW: backward walking segment. LS: left side-stepping segment. The values shown are mean (SD).

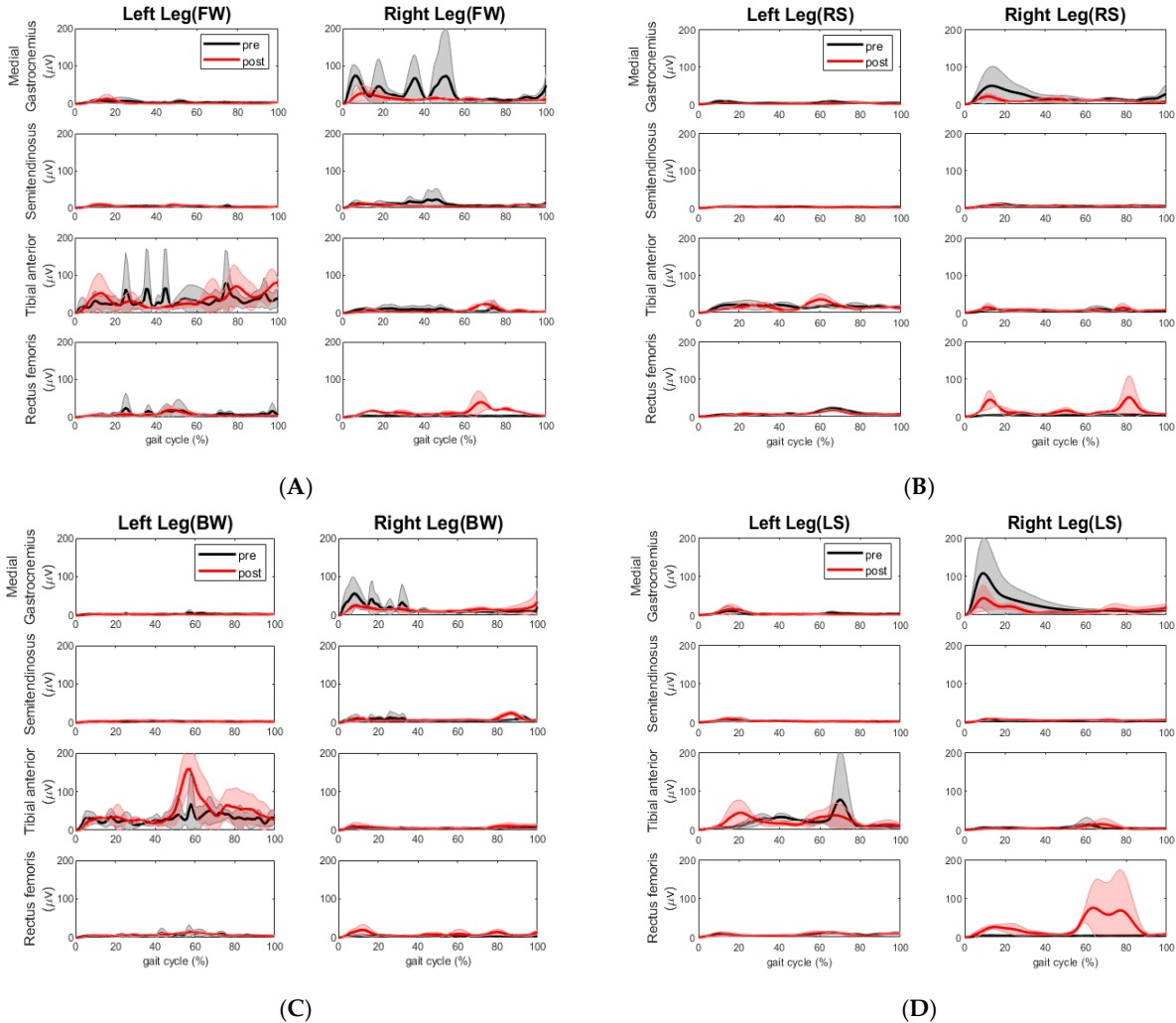

**Figure 9.** Mean EMG linear envelopes (in muV) across the gait cycle during the pre- (black) and post-training (red) tandem walking conditions for the stroke participant. The shaded regions represent $\pm 1$ SD from the mean in each walking element. (**A**) Forward walking; (**B**) Right sidestep; (**C**) Backward walking; (**D**) Left sidestep. All plots are mean values with standard deviation as the shaded region.

**Table 8.** Comparison of leg muscle activities during stance and swing phase for stroke participant.

| Gait Phase | Side | Medial Gastrocnemius (µV) | | | Semitendinosus (µV) | | | Tibial Anterior (µV) | | | Rectus Femoris (µV) | | |
|---|---|---|---|---|---|---|---|---|---|---|---|---|---|
| | | Pre | Post | *p* Value | Pre | Post | *p* Value | Pre | Post | *p* Value | Pre | Post | *p* Value |
| FW–ST | L | 14 ± 7 | 13 ± 11 | 0.887 | 8 ± 2 | 11 ± 2 | 0.161 | 96 ± 94 | 77 ± 29 | 0.702 | 40 ± 40 | 25 ± 2 | 0.446 |
| | R | 132 ± 92 | 27 ± 20 | 0.107 | 38 ± 21 | 11 ± 5 | 0.083 | 19 ± 9 | 14 ± 5 | 0.545 | 7 ± 2 | 31 ± 19 | 0.329 |
| RS–ST | L | 9 ± 2 | 4 ± 0.2 | 0.076 | 4 ± 0.4 | 5 ± 2 | 0.605 | 28 ± 2 | 35 ± 14 | 0.540 | 22 ± 1 | 15 ± 2 | 0.052 |
| | R | 50 ± 51 | 24 ± 4 | 0.463 | 9 ± 5 | 10 ± 0.7 | 0.869 | 11 ± 6 | 16 ± 7 | 0.567 | 6 ± 1 | 46 ± 23 | 0.249 |
| BW–ST | L | 10 ± 7 | 3 ± 0.4 | 0.129 | 6 ± 1 | 6 ± 0.7 | 0.894 | 90 ± 83 | 160 ± 62 | 0.229 | 22 ± 16 | 15 ± 2 | 0.435 |
| | R | 90 ± 24 | 25 ± 5 | 0.002 | 20 ± 14 | 12 ± 5 | 0.280 | 10 ± 4 | 12 ± 8 | 0.842 | 9 ± 4 | 25 ± 13 | 0.005 |
| LS–ST | L | 12 ± 11 | 16 ± 11 | 0.595 | 7 ± 6 | 9 ± 5 | 0.708 | 41 ± 5 | 56 ± 21 | 0.339 | 13 ± 3 | 14 ± 3 | 0.622 |
| | R | 114 ± 91 | 46 ± 30 | 0.185 | 7 ± 0.7 | 10 ± 2 | 0.150 | 16 ± 16 | 14 ± 7 | 0.784 | 5 ± 0.7 | 83 ± 6 | 0.158 |
| FW–SW | L | 6 ± 2 | 4 ± 2 | 0.218 | 6 ± 5 | 6 ± 3 | 0.941 | 87 ± 84 | 110 ± 20 | 0.575 | 18 ± 21 | 6 ± 0.8 | 0.261 |
| | R | 50 ± 28 | 16 ± 1 | 0.095 | 14 ± 6 | 10 ± 0.8 | 0.262 | 16 ± 19 | 27 ± 5 | 0.373 | 5 ± 1 | 43 ± 25 | 0.274 |
| RS–SW | L | 8 ± 0.5 | 5 ± 2 | 0.150 | 4 ± 0 | 3 ± 0 | 0.004 | 24 ± 0.2 | 28 ± 11 | 0.654 | 23 ± 0.3 | 16 ± 2 | 0.049 |
| | R | 31 ± 23 | 12 ± 1 | 0.290 | 7 ± 0.6 | 8 ± 1 | 0.578 | 10 ± 8 | 15 ± 9 | 0.599 | 6 ± 2 | 53 ± 55 | 0.444 |
| BW–SW | L | 7 ± 2 | 4 ± 2 | 0.081 | 4 ± 0.6 | 4 ± 0.3 | 0.342 | 77 ± 19 | 80 ± 60 | 0.947 | 15 ± 5 | 11 ± 2 | 0.178 |
| | R | 24 ± 16 | 37 ± 35 | 0.533 | 16 ± 6 | 25 ± 5 | 0.044 | 10 ± 3 | 14 ± 6 | 0.321 | 6 ± 0.8 | 17 ± 6 | 0.014 |
| LS–SW | L | 8 ± 2 | 2 ± 0.2 | 0.0180 | 3 ± 0.2 | 3 ± 1 | 0.567 | 82 ± 126 | 38 ± 30 | 0.545 | 13 ± 3 | 12 ± 1 | 0.518 |
| | R | 14 ± 2 | 20 ± 12 | 0.455 | 6 ± 0.7 | 7 ± 3 | 0.472 | 9 ± 8 | 16 ± 13 | 0.446 | 5 ± 0.3 | 84 ± 92 | 0.275 |

Note: FW: forward walking segment. RS: right side-stepping segment. BW: backward walking segment. LS: left side-stepping segment. ST: Stance phase; SW: swing phase. The values shown are mean (SD).

## 4. Discussion

Here, we conducted a case study that utilized motion analysis to investigate the effects of a 6-week multi-sensorimotor training program on two elderly participants, one of whom had suffered a chronic stroke. Following our sensorimotor training, the walking speed of the chronic stroke participant improved across all segments. The overall time it took her to complete the tandem assessment walking pattern reduced significantly by 19 s. We took this to mean a positive result for the stroke participant. We also observed more consistent steps made during post-walking assessment compared to those made during pre-assessment. The impaired left leg improved step length and cadence and the non-impaired right leg improved cadence during forward and backward walking, as shown in Table 6. The improvement of cadence and step length were also observed by the change of EMG activities. The extensor muscle on the right leg (rectus femoris) increased during forward walking. The extensor muscle activities (rectus femoris) and flexor muscle activities (semitendinosus) on the left leg decreased, whereas the same extensor muscle group on the right leg increased during right side stepping. This muscle coactivation pattern change is desired as the right leg is the leading leg for the right side stepping. During the backward walking, the increased extensor and decreased flexor muscle activities on the right leg may have allowed this stroke participant to have better control of footwork. During the left sidestepping, the increased extensor (rectus femoris) and flexor muscle (semitendinosus) activities on the right leg indicated that this participant relied on the right leg to make strides, although the impaired left leg was the lead leg. However, the decreased flexor muscle (medial gastrocnemius) activities might be a sign of reduced impairment in the left leg. Surely, it would be better if there was increased extensor muscle activity from the left leg and a more coordinated extensor and flexor muscle coactivation pattern during all walking segments.

The healthy participant displayed a more flexed hip and knee on the right side across all walking sections post assessment, as shown in Figure 6. Although there is no significant improvement in walking time and step length during post assessment, the training may have led to the more balanced walking strategy in which the healthy participant walked along the path with reduced speed but better compliance of "heel-to-toe" tandem forward and backward walking. The more balanced walking strategy is also indicated by less difference between the cadences of the two legs. During the pre-assessment, the cadence of his right leg was higher than that of his left leg for forward and the right side stepping, whereas the cadence of the right leg was lower than that of the left leg for backward and left side stepping. In comparison, during the post-assessment, the cadence of his left leg

increased during the forward walking section and decreased during the other three sections, whereas the cadence of his right leg increased during the backward walking section and decreased during the other three sections. Analysis of his EMG activity provided evidence for this shift. This finding might be due to the fact that the healthy elderly participant used a different side of leg to lead the forward (left leg lead) and backward (right leg lead) walking during post evaluation. We noticed that the healthy participant started from point A in the pre-assessment trial but unintentionally started from point D in the post-assessment trial. However, both healthy and stroke participants followed the same walking order.

In terms of joint angle, for the stroke participant, her joint ROMs during forward walking did not improve to the level of the reported minimally clinically important difference (MCID) from the straight-line walking study by Guzik et al. [22,23]. Her impaired left leg had a significant increase in hip joint ROM ($5.8 \pm 2.4$ deg, $p = 0.019$) during backward walking and a marginally significant increase in knee joint ROM ($9.0 \pm 6.7$ deg, $p = 0.062$) during the left side stepping. Her right hip joint ROM increased significantly ($4.1 \pm 3.7$ deg, $p = 0.046$) and the right knee joint ROM increased significantly ($8.1 \pm 2.9$ deg, $p = 0.007$) during backward walking. For the healthy participant, the significant increases were only found in his right knee joint ROM ($4.2 \pm 3.0$ deg, $p = 0.031$) and left ankle joint ROM ($5.5 \pm 0.4$ deg, $p = 0.006$) during the left side stepping. Currently, we did not find any studies that explicitly report MCIDs of backward walking and side stepping. However, the studies on walking strategies reported that backward walking involves conscious extension of the hip joint extension which is different from forward walking. The conscious extension of the hip joint during backward walking is difficult not only for hemiplegia but also for healthy elderly individuals due to the sensory conflicts. It has been reported that peak hip joint extension and peak knee joint flexion, ankle joint plantar flexion movement range, walking speed and cadence are significantly lower in backward walking than in forward walking for hemiplegia and healthy individuals. As for the side stepping, Sparto et al. reported two different strategies of lateral step behaviors to initiate stepping in older adults based on the profile of vertical ground reaction force (VGRF): one strategy involving a single postural adjustment (PA) made to directly unload the stepping leg which is adopted by younger adults; another one involving two PAs prior to liftoff, with the stepping leg initially loading to propel the center of mass toward the stance leg, followed by the VGRF decreasing to unload the stepping leg so that the step could be taken [24]. Two PAs to initiate stepping may be caused by aging related declines in hip abductor muscle composition and torque generation [25]. It was found that participants with a preferred strategy of generating two PAs had greater frequency of fall than those with a preferred strategy of generating one PA [24]. Borrelli et al. reported the similar findings that older adults are prone to multistep responses when exposed to lateral pull at waist level, which are associated with an increased fall risk [26]. We argue that the survivor of stroke with hemiplegia will tend to use the preferred strategy of generating two PAs, especially when the impaired leg initiates the stepping.

Overall, the results show that these dependent variables, walking speed, step length and cadence in spatiotemporal measures, and knee and hip joint angles are more sensitive to gait improvement or learning effect, among all the dependent variables. The walking path chosen for this gait evaluation involved the participants in four types of walking exercises (forward, backward, and left- and right-side stepping). This path provided us evidence on how well the participant performed a set of walking exercises in a row and the transition in between in terms of joint kinematics, muscle activities and postural stability. Compared to an only-straight-line walking path, this multidirectional path would better indicate participants' walking ability in their daily activities. Because the results shown here were limited to two participants (one healthy elderly and one stroke with hemiplegia), it is fairly early to say if the change of joint angle profile, spatiotemporal measures, as well as EMG activities is a common trend for each population.

Although this study focuses on the biomechanical analysis of walking, we had used two clinical metrics such as the balance error scoring system (BESS) and activities specific

balance confidence (ABC) [26] to evaluate the balance control. In the BESS assessment, the participants perform double-leg, single-leg and tandem stances as they stand on either hand or foam support surfaces with eyes closed [26]. The BESS assesses the trunk deviation from its upright posture and the number of deviations are counted as errors for six 20-s trials per condition [26]. The ABC assesses the participant's balance confidence using a list of survey questions relating to the gait and balance [26]. A score of 100 represents high balance level and a score between 50 and 80 represents a moderate balance level. The low balance level has an ABC score of less than 50. Some of the clinical metrics currently used are Fugl-Meyer Assessment (FMA) for Lower Extremity, National Institutes of Health Stroke Scale (NIHSS), Timed Up and Go Test (TUG) [27] and Tinetti Gait and Balance Assessment Tool [28]. The FMA Lower Extremity [29] is used to assess the level of lower limb impairment by quantifying the joint movement in supine, sitting or standing positions. The assessment of hip, knee and ankle flexion extension under these static positions may not reflect well the capability of these joints during the actual walking. The NIHSS is a systematic assessment tool that provides a quantitative measure of stroke-related neurologic deficit [30]. Only item 6 of NIHSS assesses the motor function related to the leg in a supine position. TUG is a Falls risk screening tools for falls among older adults. However, it has been reported that the TUG should no longer be used as a falls risk assessment in community-dwelling elderly people due to its limited predictive ability [27]. Tinetti assessment covers both gait and balance. However, the scoring of performance in each task is based on the examiner's subjective observation and also, its binary scoring metrics is rather rigid to indicate the progress.

Sensorimotor exercises were developed and are of interest because, if successful in improving balance, they can be eventually translated to home-based practices and training. With the aging population in the United States and worldwide rapidly increasing, our society is in great need of safe and effective solutions that can be easily accessed on a daily basis by people regardless of their socio-economic status [31,32]. Due to the pandemic and the evolution of technology, lifestyles are shifting to make daily activities gravitate more towards the home and local community. To meet this need and shift, exercise programs that can be implemented outside of the laboratory or hospital settings have been developed and studied for fall prevention practices [33–35]. These programs reportedly reduced the rate of falls by improving balance, leg strength and function, and physical activity and balance confidence in older adults living in the community [36,37]. The benefits of these exercises programs and sensorimotor therapies have led us to develop a multi-sensorimotor training program that can be delivered to meet both needs of gait rehabilitation and fall prevention in home environment.

The present assessment protocol has two novelties. First, a rectangle walking path was used to assess the movement and balance. This path requires participants to follow the path and perform various walking patterns (tandem forward, side-stepping led by right leg, tandem backward, side-stepping led by left leg) and transitions in between. Performing these tasks in sequence and in short distance can assess the change of dynamic balancing and multisensory integration in different walking situations, which is critical for fall preventions. Second, we developed a data-processing pipeline that can produce spatiotemporal measures, the joint angle profile muscle activity profile with respect to the gait cycle, and can perform the statistical analysis.

Regarding the limitations of this study, first, the current training intensity of our 6-week SMT may not be sufficient for both participants to have clinically significant improvement in joint range of motion at each walking section. We plan to increase the training intensity from 30 min to 45 min and from 2 sessions a week to 3 sessions a week. The walking distance is relatively short on each walking segment as shown in Figure 1C. Both participants may be not able to reach the steady-state walking pace, which may compromise the assessment of their movement and balance. Secondly, our motion capture system did not have the force plate to collect the ground reaction force (GRF) data which can provide more accurate time information on the gait events such as heel strike and toe off. The GRF data can

also verify the surface EMG data we collected from four leg muscles, because the external moment calculated by GRF data should be balanced by internal muscle forces (internal moment). We therefore plan to install the force plate to collect GRF data. In addition, the current assessment protocol did not include the standardized gait assessment tasks. As we continue the gait analysis based on motion capture in more participants, we plan to include clinical assessments (e.g., ten-meter walk test (10MWT) to assess walking speed, six-minute walk test (6MWT)) to assess aerobic capacity and endurance [38].

## 5. Conclusions

Our assessment protocol uses the motion capture system with wireless surface electromyography sensors to assess the effectiveness of a sensorimotor training on improving the gait and balance of elderly individuals. The assessment focuses on the joint kinematics and muscle activities in lower extremities as well as the spatiotemporal measures of the walking, which allows objective and longitudinal comparison. A data-processing pipeline was implemented to process the kinematic data and muscle activity of one healthy and one stroke participants. The healthy control participant had minimum improvement in joint ROM and EMG activity during post assessment. This is expected, as the tasks might not be challenging enough to induce the motor learning effect. Among the selected outcome variables, the stroke participant showed a greatly reduced task completion time, increased step length in the left impaired leg and increased cadence in both legs during forward and backward walking. We found that the sensorimotor training also improved the muscle activity to some extent. The extensor muscle (rectus femoris) had increased EMG activity in the right leg through all the walking sections. The activities of both flexor muscles (semitendinosus and medial gastrocnemius) on the left leg decreased at certain walking segments. However, most joint angles in the impaired left leg did not show the improvement except the hip joint ROM during the backward walking and knee joint ROM during the left side stepping. This analysis serves an initial step for us to conduct a large-scale biomechanics study on the effects of our sensorimotor training on both elderly and stroke groups. By leveraging our assessment protocol, we can assess an individual's gait more efficiently and thus develop a training plan customized to their needs.

**Author Contributions:** Conceptualization, J.C. and L.A.T.; methodology, J.C. and L.A.T.; software, J.C. and R.R.; validation, J.C. and L.A.T.; formal analysis, J.C. and R.R.; investigation, J.C., R.R. and L.A.T.; resources, L.A.T.; data curation, J.C.; writing—original draft preparation, J.C. and L.A.T.; funding acquisition, L.A.T. All authors have read and agreed to the published version of the manuscript.

**Funding:** This work was supported by NIH 1R25AG067896, NSF #1533479, 1654474, and 1700219, DACL MOU, and NASA #80NSSC21K2061.

**Institutional Review Board Statement:** The study was conducted in accordance with the Declaration of Helsinki and approved by the Institutional Review Board of University of the District of Columbia (protocol code: #979744).

**Informed Consent Statement:** Informed consent was obtained from all subjects involved in the study.

**Data Availability Statement:** The data presented in this study are available on request from the corresponding author.

**Conflicts of Interest:** The authors declare no conflict of interest.

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
