# Peer review of "Motion Analysis of Balance Pre and Post Sensorimotor Exercises to Enhance Elderly Mobility: A Case Study"

_applsci, doi:10.3390/app13020889_

Round 1
Reviewer 1 Report (Previous Reviewer 2)
The authors measured two subject (one healthy and one patient). Please add to title of article, it is a cae study. Recommended title is
Motion analysis of balance pre and post sensorimotor 2 exercises to enhance elderly mobility - a case study
Please emphasize that in the aim and in the discussion too.
Author Response
Review 1:
The authors measured two subject (one healthy and one patient). Please add to title of article, it is a case study. Recommended title is Motion analysis of balance pre and post sensorimotor 2 exercises to enhance elderly mobility - a case study
The title has changed to Motion analysis of balance pre and post sensorimotor exercises to enhance elderly mobility - a case study.
Please emphasize that in the aim and in the discussion too.
The objective of study described by the manuscript title has been emphasized in the introduction of the manuscript shown as below. This manuscript aims to explain our assessment protocol and, as a case study, to describe motion analysis that investigates the effects of our sensorimotor training on the mobility of two elderly participants (one with chronic stroke). (page 2 lines 91 – 93)
This objective has also been emphasized in the discussion section of the manuscript shown as below. Here we conducted a case study that utilized motion analysis to investigate the effects of a 6-week multi-sensorimotor training program on two elderly participants, one of which suffers from chronic stroke. (page 16 lines 363 – 365)

Reviewer 2 Report (Previous Reviewer 3)
This research found that the sensory-motor training for 6 weeks in the storke patient, resulted in the improvement of their gait abilities. It is suitable for publication for this journal as a brief report, however, the authors should modify some parts. Please comfirm the list below.
Minor comments
L 213-215 The font is different from the other parts.
Figure 5 The "Visual3" should be changed to "Visual 3D".
Figure 5 The bottom of the word "MATAB" is missing.
I have recommended that this manuscript shouled be submitted as brief report because of the shortage of sample size. The authors have done as pointed out, so the rivision is enough. However, there are several concerns about the main part and intruroduction of this paper.
The authors concluded that the gait ability has improved due to SMT. Sensory functions are essential for the motor control including the gait, however, these mechanism are extremely complicated. Not only sesory function are affected the performance but also the musculoskeletal and central nerve system. In addition, the connection between them are also important. There is no mention about these factors in the introduction part.
I think that the improvement resulted in the overall of the training but not only the sensory function. The authors shouled consider the effects of overall aspects for input, integrate, and output, and explain those in the introduction.
Author Response
Reviewer 2:
Minor comments
L 213-215 The font is different from the other parts.
The font of texts in lines 213-215 has been updated to match the rest of manuscript.
Figure 5 The "Visual3" should be changed to "Visual 3D".
We have changed Visual3 to Visual 3D in Figure 5.
Figure 5 The bottom of the word "MATLAB" is missing.
The bottom of the word “MATLAB” is fixed now.
I have recommended that this manuscript should be submitted as brief report because of the shortage of sample size. The authors have done as pointed out, so the revision is enough. However, there are several concerns about the main part and introduction of this paper.
The authors concluded that the gait ability has improved due to SMT. Sensory functions are essential for the motor control including the gait, however, these mechanisms are extremely complicated. Not only sensory function can affect the performance but also the musculoskeletal and central nerve system. In addition, the connection between them are also important. There is no mention about these factors in the introduction part.
I think that the improvement resulted in the overall of the training but not only the sensory function. The authors should consider the effects of overall aspects for input, integrate, and output, and explain those in the introduction.
We have taken the reviewer’s suggestion and have added the following paragraph in the introduction.
Ideally the design of SMT should aim to facilitate not only the sensory input, but also integration of information in central nervous system and motor output. In particular, a few functions and systems need to be intact in order for a person to have normal gait. These functions and systems include locomotor function which initiates and sustains rhythmic gait, postural reflex, sensory function, sensory integration, the musculoskeletal system and cardiopulmonary functions. The tasks in our SMT focus on the proprioceptive stimulations, locomotor function and the musculoskeletal system by asking participants to perform the standing, step-up, and walking tasks on different surfaces with stable posture through three stages. (page 2 lines 69 – 77)

This manuscript is a resubmission of an earlier submission. The following is a list of the peer review reports and author responses from that submission.
Round 1
Reviewer 1 Report
The paper :”Biomechanical analysis of balance pre and post sensorimotor exercises to enhance elderly mobility” from Chen et al. describes a 6-week sensorimotor training program that aims to improve balance and walking to prevent falls.
Unfortunately, the introduction to the abstract is a bit lengthy ("poor gait can cause frequent tripping and falls ....") but also a bit confusing for the reader. It is spoken by two people, but then also by "both participant groups". In the text, the actual implementation of the training program becomes clear. There are two people being examined.
The introduction of the paper is also somewhat general and rambling. The "Frugal-Meyer Assessment (FMA)" test certainly means the Fugl-Meyer Assessment of Motor Recovery. Such errors should not appear in the manuscript.
Methodologically, the presented question about the impact of sensorimotor training on fall prevention cannot be answered. The study design with a single test person (patient) and a single control person as a control group is not sufficient.
The biomechanical methods used may offer sufficient accuracy in everyday clinical practice, which I cannot judge here, but statements on the possible effects of a training protocol are completely impossible in this context. This is especially true if you want to make comparisons with conventional scales.
Basically, the “biomechanical approach” to reviewing training programs is a useful way of objectifying. I would therefore like to encourage the authors to continue on this path. An in-depth study of the theoretical background (multi-sensory integration) is also strongly recommended.
Reviewer 2 Report
Abstract:
Please provide the age with mean and SD
Please provide numerical data about the joint angles.
Introduction:
Please provide exactly the gap of the previous research.
Please provide exactly the types of walking.
Materials and Methods
Please provide all antropometric data (age, mass, height) with mean and SD
If you used kg, you should write mass (weight is a force and you should use N)
Please provide the velocity of motion. Was it self-selected or constant?
Please provide all calculated parameters in a Table.
Please summarize the measurement protocol in a flowchart (which motions are measured, how was the order)
Please provide few details about the sensorimotor training.
Results
How did you determine the p values? Which analysis did you use?.
Discussion
Please use motion analysis instead of biomechanical metrics
Please provide exactly the limitations of the present study.
Please provide exactly the novelty of the present research
Reviewer 3 Report
The participants of this study was only two, so it should be submitted as a short note or brief communication.
I think the authors should change the manuscript type, and then they should resubmit it